# TMEM16A Maintains Acrosomal Integrity Through ERK1/2, RhoA, and Actin Cytoskeleton During Capacitation

**DOI:** 10.3390/ijms26083750

**Published:** 2025-04-16

**Authors:** Ana L. Roa-Espitia, Tania Reyes-Miguel, Monica L. Salgado-Lucio, Joaquín Cordero-Martínez, Dennis Tafoya-Domínguez, Enrique O. Hernández-González

**Affiliations:** 1Department of Cell Biology, Center of Research and Advanced Studies of the National Polytechnic Institute, México Av. Instituto Politécnico Nacional 2508, México City 07360, Mexico; aroa@cinvestav.mx (A.L.R.-E.); tania.reyes@cinvestav.mx (T.R.-M.); monisalgado@cinvestav.mx (M.L.S.-L.); leslie.tafoya@cinvestav.mx (D.T.-D.); 2Department of Health Sciences, Division of Biological and Health Sciences, Universidad Autónoma Metropolitana, Unidad Iztapalapa, Av. San Rafael Atlixco No. 186, Colonia Vicentina, Alcaldía Iztapalapa, México City 09310, Mexico; 3Department of Biochemistry, National School of Biological Sciences, National Polytechnic Institute, Prolongación Manuel Carpio y Plan de Ayala s/n Col, Santo Tomás, Del. Miguel Hidalgo, México City 07738, Mexico; jcorderom@ipn.mx

**Keywords:** sperm, fertilization, chloride channels, bicarbonate influx, acrosome structure

## Abstract

Mammalian spermatozoa undergo a series of physiological and biochemical changes in the oviduct that lead them to acquire the ability to fertilize eggs. During their transit in the oviduct, spermatozoa face a series of environmental changes that can affect sperm viability. A series of ion channels and transporters, as well as the sperm cytoskeleton, allow spermatozoa to remain viable and functional. Cl^−^ channels such as TMEM16A (calcium-activated chloride channel), CFTR (cystic fibrosis transmembrane conductance regulator), and ClC3 (chloride voltage-gated channel 3) are some of the ion transporters involved in maintaining cellular homeostasis. They are expressed in mammalian spermatozoa and are associated with capacitation, acrosomal reaction, and motility. However, little is known about their role in maintaining sperm volume. Therefore, this study aimed to determine the mechanism through which TMEM16A maintains sperm volume during capacitation. The effects of TMEM16A were compared to those of CFTR and ClC3. Spermatozoa were capacitated in the presence of specific TMEM16A, CFTR, and ClC3 inhibitors, and the results showed that only TMEM16A inhibition increased acrosomal volume, leading to changes within the acrosome. Similarly, only TMEM16A inhibition prevented actin polymerization during capacitation. Further analysis showed that TMEM16A inhibition also prevented ERK1/2 and RhoA activation. On the other hand, TMEM16A and CFTR inhibition affected both capacitation and spontaneous acrosomal reaction, whereas ClC3 inhibition only affected the spontaneous acrosomal reaction. In conclusion, during capacitation, TMEM16A activity regulates acrosomal structure through actin polymerization and by regulating ERK1/2 and RhoA activities.

## 1. Introduction

Recently ejaculated spermatozoa are incapable of fertilizing eggs; they acquire this ability as they travel through the female’s reproductive tract, undergoing a series of physiological and biochemical changes. Austin and Chang introduced the term “capacitation” to describe the changes spermatozoa must undergo before they can pass through the acrosomal reaction (AR) and fertilize the eggs. The term capacitation is currently defined as a series of physiological and biochemical changes that spermatozoa experience that enable them to carry out AR, endow them with hyperactivated motility, and prepare them to fertilize eggs [1].

One necessary process that occurs during sperm capacitation is actin cytoskeleton remodeling, which implies an increase in actin polymerization. This process is crucial as its absence significantly reduces the spermatozoa’s ability to fertilize eggs [2,3,4,5]. In mammalian spermatozoa, actin polymerizes mainly in the acrosome, the flagellum, and the equatorial segment [5,6]. In the acrosomal region, a mesh is formed on the cytoplasmic side of the plasma membrane that prevents spermatozoa from undergoing spontaneous RA [5,7,8]. It was recently shown that Rho family proteins such as Cdc42, Rac1, and RhoA regulate actin polymerization [6,7,9,10], although little is known about the mechanisms that regulate the activities of these Rho proteins.

Various ions, such as Ca^2+^, HCO_3_^−^, Na^+^, K^+^, and Cl^−^, are necessary for capacitation, acrosomal reaction, and motility [11,12]. Three central Cl^−^ channels have been described in mammalian spermatozoa: Ca^2+^-activated Cl^−^ channels (TMEM16A) [13,14], the regulator of conductance cystic fibrosis transmembrane channel (CFTR) [15,16], and chloride channel 3 (ClC3) [17,18]. These channels are associated with different physiological processes in spermatozoa, including capacitation, acrosomal reaction, and motility [13,14,16,17,18]. In human and guinea pig spermatozoa, TMEM16A is associated with capacitation, acrosomal reaction, and motility through the regulation of Cl^−^ influx since the inhibition of TMEM16A during capacitation causes an increase in intracellular Cl^−^ concentration, which, in turn, decreases the intracellular concentration of Ca^2+^, leading to blockage of capacitation, AR, and a decrease in motility [13,14]. CFTR is necessary for the influx of Cl^−^ and HCO_3_^−^ that occurs during capacitation; therefore, inhibiting this channel prevents spermatozoa from undergoing normal plasma membrane hyperpolarization, ZP3-induced RA, and hypermotility. These events are associated with the ability of spermatozoa to fertilize eggs [15,16]. ClC3′s activity is implicated in motility and cell volume regulation in human spermatozoa [18].

TMEM16A, CFTR, and ClC3 activities have also been associated with the actin cytoskeleton. TMEM16A interacts indirectly with the actin cytoskeleton through the ERM scaffold proteins (ezrin, radixin, and moesin), which organize the actin cytoskeleton by binding actin filaments to the plasma membrane [19]. The relationship between TMEM16A and the actin cytoskeleton goes beyond a simple interaction; TMEM16A expression is essential for the structuring of the actin cytoskeleton, and blocking it reduces the number of actin filaments induced by oxytocin in smooth muscle cells [20]. Likewise, TMEM16A overexpression enhances Angiotensin II-induced RhoA/ROCK2 activation through FAK phosphorylation [21]. TMEM16A activity is required for the activation of several signaling pathways, including CaMII/AKT/SRC and Ras/Raf/MEK, both of which are involved in the activation of ERK1/2 [22]. It was recently reported that ERK1/2 is involved in the activation of GEF-H1, a fundamental step in RhoA activation [10]. CFTR also influences the organization of the subcortical actin cytoskeleton [23]. The interaction between CFTR and the actin cytoskeleton is indirect and involves scaffolding proteins such as NHERF1 and ezrin. Additionally, CFTR forms a multiprotein complex with NHERF1/Ezrin/RhoA/ROCK that directly influences the stability of the actin cytoskeleton; thus, alterations in CFTR or NHERF1 destabilize the actin cytoskeleton [24]. The activity of ClC3 is modulated by its direct interaction with the subcortical actin cytoskeleton [25].

The journey of spermatozoa from the testicles to the eggs involves passage through different environments with different osmotic compositions. To overcome this adversity and control cell volume, mammalian spermatozoa use different mechanisms to adapt to these environments, among which potassium and chloride channels have an essential role [26,27]. Maintaining homeostasis of intracellular Cl^−^ concentration ([Cl^−^]i) is required for the regulation of cell volume. Changes in [Cl^−^]i cause important changes in cell volume since the regulation of [Cl^−^]i is vital to maintaining ionic homeostasis and water concentration; changes in [Cl^−^]i cause water movements and, therefore, changes in the cell volume [28]. [Cl^−^]i increases under normal capacitation conditions, a change that has been reported for mouse and guinea pig spermatozoa [14,15]. In this regard, chloride channels such as TMEM16F, CFTR, ClC2, and ClC3 play an essential role in the regulation of cellular volume [28]. However, little is known about whether TMEM16A is involved in cell volume regulation; therefore, this study aimed to determine whether TMEM16A activity is essential for maintaining spermatozoa integrity, whether the actin cytoskeleton is involved in regulating cellular integrity, and whether TMEM16A activity is required for structuring the actin cytoskeleton during capacitation.

## 2. Results

### 2.1. The Tested Inhibitors Do Not Alter Sperm Viability

First, TMEM16A, CFTR, and ClC3 inhibitors were assessed for their effects on sperm viability. Spermatozoa non-capacitated and spermatozoa capacitated in the presence or absence of T16Ainh-A01 (1 µM), CFTR-inh (10 nM), or NPPB (25 µM) showed significantly low percentages of propidium iodide staining compared with the control spermatozoa treated with Triton X100, as well as percentages similar to those exhibited by spermatozoa non-capacitated or capacitated in the absence of inhibitors (Figure 1). These results suggest that the inhibitors tested in the current study do not alter sperm viability.

### 2.2. TMEM16A Inhibition Disrupts the Acrosome Structure

To determine whether the inhibition of TMEM16A, CFTR, or ClC3 affects sperm integrity, spermatozoa were capacitated in the absence or presence of T16Ainh (1 µM), CFTRinh (10 nM), or NPPB (25 µM), and sperm morphology was analyzed. The analysis showed a single, discrete change in acrosome size of spermatozoa capacitated in the presence of TMEM16A (Figure 2A). To corroborate this observation, the acrosome area of spermatozoa capacitated in the presence or absence of the inhibitors above was assessed. The spermatozoa did not show changes in acrosome area compared to non-capacitated spermatozoa after 60 min of capacitation in the absence of the inhibitors (Figure 2B). Similar results were observed when spermatozoa were capacitated in the presence of CFTRinh or NPBB (Figure 2B). However, spermatozoa capacitated in the presence of T16Ainh showed a significant increase in the acrosome area after 10 min of incubation (Figure 2B), when approximately 90% of sperm maintain their acrosome (Appendix A).

The above results indicate that TMEM16A inhibition primarily disrupts acrosome integrity. To test this hypothesis, the localization of calreticulin (CRT), a sperm protein located in the acrosomal matrix [29,30], was determined. Both non-capacitated and capacitated spermatozoa showed strong fluorescence exclusively in the acrosomal region (Figure 3, top and middle panels). Spermatozoa capacitated in the presence of T16Ainh showed three patterns of CRT localization: (1) a pattern with dim fluorescence dispersed throughout the acrosome, (2) a pattern with fluorescence concentrated in the apical region of the acrosome, and (3) a fluorescence pattern similar to that of capacitated spermatozoa (Figure 3, bottom panel). Taken together, these results indicate that TMEM16A inhibition affects acrosome integrity.

### 2.3. Inhibition of TMEM16A Prevents Actin Polymerization

The importance of the actin cytoskeleton in the stability of different cellular structures, including the sperm acrosome, is well known. The actin cytoskeleton prevents these cellular structures from undergoing damage [31,32]. Given the relationship between TMEM16A and the actin cytoskeleton [19], this study assessed whether the inhibition of TMEM16A affects the actin polymerization that occurs during capacitation. Spermatozoa were capacitated for 60 min in the absence or presence of T16Ainh, and F-actin was detected and measured using phalloidin-FITC. Non-capacitated spermatozoa (0 min of incubation) showed a dim fluorescence in both the acrosomal region and the flagellum, especially in the midpiece region (Figure 4A). In contrast, capacitated spermatozoa in the absence of any inhibitor showed a significant increase in fluorescence compared to non-capacitated spermatozoa (Figure 4A,B). This increase was abrogated when spermatozoa were capacitated in the presence of T16Ainh (Figure 4A,B). The effect of TMEM16A inhibition on actin polymerization differed from that of CFTR or ClC3 inhibition during capacitation. In both cases, spermatozoa capacitated in the presence of CFTRinh or NPPB increased the fluorescence produced by phalloidin-FITC in a manner similar to that shown by capacitated spermatozoa (Figure 4A,B). These results suggest that only TMEM16A activity is associated with cytoskeletal remodeling during capacitation.

### 2.4. TMEM16A, CFTR, and ClC3 Inhibition Alters Intracellular Cl^−^ Homeostasis and the Intracellular pH

TMEM16A is involved in [Cl^−^]i and pHi homeostasis in spermatozoa; therefore, the effects of TMEM16A inhibition on [Cl^−^]i and pHi were assessed and compared with the effect of CFTRinh and NPPB.

#### 2.4.1. Intracellular Concentration of Cl^−^

[Cl^−^]i increases during capacitation [14,15]. The [Cl^−^]i measurement assays performed in this study confirm our previous observations. Similarly, [Cl^−^]i increased significantly in spermatozoa capacitated in the presence of T16inh, CFTRinh, or NPPB compared with spermatozoa capacitated in the absence of these inhibitors (Figure 5A). In the case of CFTR inhibition, the increase in [Cl^−^]i was also significantly greater than when TMEM16A or ClC3 was inhibited (70.9 ± 1.7 mM vs. 55.8 ± 3.5 mM or 53.1 ± 3.6 mM. S.E. *n* = 5, *p* < 0.05).

#### 2.4.2. Intracellular pH

Capacitation is dependent on the influx of bicarbonate (HCO_3_^−^) [27], which is closely associated with Cl^−^ fluxes [33,34,35]. Given that the inhibition of TMEM16A prevents Cl^−^ fluxes during capacitation [13,14], we assessed the pHi to investigate whether the HCO_3_^−^ influx was also affected. A significant increase in pHi was observed in capacitated spermatozoa compared with non-capacitated spermatozoa. This increase was not observed when the spermatozoa were capacitated in the presence of T16Ainh (Figure 5B). In contrast, CFTRinh and NPPB significantly decreased intracellular pH compared with T16Ainh (6.6 ± 0.10 and 6.6 ± 0.13 vs. 7.44 ± 0.10, S.E., *p* < 0.05, *n* = 5). These results suggest that Cl^−^ fluxes are essential for maintaining the pHi homeostasis. However, these changes may not be associated with actin polymerization.

### 2.5. TMEM16A Inhibition Prevents ERK1/2 and RhoA Activation

TMEM16A activity is reportedly associated with the activation of the ERK1/2 pathway [22]; recently, ERK1/2 has also been reported to be associated with RhoA activation [10]. Therefore, assays were performed to determine whether TMEM16A inhibition during capacitation affects ERK1/2 and RhoA activity. Spermatozoa were capacitated in the absence or presence of T16Ainh, and ERK1/2 phosphorylation at Tyr 204 (p-ERK1/2) was revealed and assessed by Western blotting. Capacitated spermatozoa (10 min, time at which ERK1/2 reaches its maximum activity [10]) showed a significantly higher level of phosphorylated ERK1/2 than non-capacitated spermatozoa (Figure 6A,B), while spermatozoa capacitated in the presence of T16Ainh showed levels of ERK1/2 phosphorylation similar to those of non-capacitated spermatozoa (Figure 6A,B).

To determine whether TMEM16a activity is also involved in RhoA activity and, thus, in actin polymerization, spermatozoa capacitated in the absence or presence of T16Ainh were used to isolate RhoA-GTP using a pull-down assay based on Rhotekin-RBD. When the isolated Rho-GTP protein was visualized using Wb, a significantly higher quantity of Rho-GTP was identified in capacitated spermatozoa than in non-capacitated spermatozoa. This increase in Rho-GTP was not observed in spermatozoa capacitated in the presence of T16Ainh (Figure 6C,D). To confirm that ERK1/2 is involved in the signaling pathway that regulates RhoA, FR180204, a specific inhibitor of ERK1/2, was also tested in the pull-down assay. The results showed that the quantity of RhoA-GTP isolated from capacitated spermatozoa in the presence of FR180204 was similar to that obtained from non-capacitated spermatozoa. A control assay performed with capacitated spermatozoa in the presence of C3, a specific inhibitor of RhoA, also showed levels similar to those of non-capacitated spermatozoa (Figure 6C,D).

To confirm that ERK1/2 inhibition prevents actin polymerization [10], phalloidin-FITC fluorescence levels were assessed in capacitated spermatozoa in the presence of FR180204, and values similar to those of non-capacitated spermatozoa were obtained (Figure 4A,B). Overall, these results indicate that TMEM16 is involved in the regulation of the actin cytoskeleton through ERK1/2 and RhoA. Therefore, inhibition of TMEM16A affects the organization of the actin cytoskeleton and, consequently, the volume and structure of the acrosome.

### 2.6. TMEM16A, CFTR, and ClC3 Inhibition Inhibit the Normal Course of Capacitation and Spontaneous Acrosomal Reaction

To define whether independent inhibition of TMEM16A, CFTR, and ClC3 has the same effect on capacitation and spontaneous acrosomal reaction (sAR), spermatozoa were capacitated (60 min) in the absence or presence of specific inhibitors of TMEM16A (1 µM T16inh), CFTR (10 nM CFTRinh), or ClC3 (25 µM NPPB). The capacitation and acrosomal reaction states were determined using the CTC technique. Two patterns were evaluated: pattern B, representing those spermatozoa that experienced capacitation, and pattern AR, representing those spermatozoa that underwent sAR [36]. The control, spermatozoa capacitated in the absence of inhibitors, showed a significant increase in pattern B compared with non-capacitated spermatozoa (Figure 7A). Spermatozoa capacitated in the presence of T16Ainh or CFTRinh had significantly lower pattern B levels compared with the control (capacitated spermatozoa in the absence of any inhibitors); however, these were not significantly different from pattern B values in non-capacitated spermatozoa (Figure 7A). Notably, in the case of ClC3 inhibition by NPPB, the pattern B level was lower than in the control. This decrease in pattern B was not significant compared with the control but was significant compared with non-capacitated spermatozoa. (35.2 ± 2.7% vs. 14.0 ± 2.3%, *n* = 3, *p* < 0.05).

Regarding the pattern AR, the control showed a significant increase compared with non-capacitated spermatozoa (Figure 7B). Spermatozoa capacitated in the presence of T16Ainh, CFTRinh, or NPPB blocked—to different degrees—the increase in the pattern AR compared with spermatozoa capacitated in the absence of the inhibitors (Figure 7B). The analysis also revealed a significant increase in AR pattern values of capacitated spermatozoa in the presence of NPBB compared to non-capacitated spermatozoa (34.89 ± 3.5% vs. 10.33 ± 2.6%, *n* = 3, *p* < 0.05). These results highlight the effects of TMEM16A, CFTR, and ClC3 on sAR.

## 3. Discussion

In the present study, we have revealed important aspects of the function of the Ca^2+^-dependent Cl^−^ channel—called TMEM16A or Ano1—in sperm physiology that differentiate it from other Cl^−^ channels, such as CFTR and ClC3. The results show that TMEM16A plays a relevant role in maintaining the structure of the acrosome by regulating actin polymerization by controlling the activity of two essential elements of capacitation: ERK1/2 and RhoA.

TMEM16A activity is associated with the regulation of [Cl^−^]i and decreased secretory volume due to its Cl^−^ influx activity [28]. However, little is known about the relationship between TMEM16A and cell volume regulation. Our data show that inhibiting TMEM16A alters the acrosome structure without compromising sperm viability, as indicated by the viability results (Figure 1). Morphological analysis shows that only the acrosomal region is altered by TMEM16A inhibition, an effect that was not observed when CFTR and ClC3 were inhibited (Figure 2B). A key element of volume regulation in spermatozoa, and especially in maintaining the structural integrity of the acrosome, is the mesh of actin filaments produced during capacitation [7,31,37]. Before capacitation, the structure of the acrosomal region, especially that of the acrosome, is maintained by the cytoskeleton formed by spectrin, preventing spermatozoa from undergoing spontaneous acrosomal reaction. During capacitation, the spectrin cytoskeleton is degraded by calpain [38] and its function is replaced by the actin cytoskeleton that is structured in the apical region of the acrosome between the plasma membrane and the external acrosomal membrane [5,6,7]. We, therefore, suggest that an important function of TMEM16A is to maintain sperm volume, especially of the acrosome, by regulating the structuring of the actin cytoskeleton, which develops between plasma and external acrosomal membranes during capacitation. Other sperm structures are possibly not affected by the inhibition of TMEM16A since they are structured by cytoskeletal elements other than actin filaments, such as the flagellum formed by the axoneme and dense fibers or the nucleus surrounded by the perinuclear theca, which is a cytoskeleton formed by proteins such as calylicin, soluble alkaline proteins, or capping proteins α3 and β3 [39,40]. On the other hand, our results show that inhibition of CFTR and ClC3 did not affect acrosome structure or actin polymerization, suggesting that TMEM16A activity has a major role in regulating acrosome structure.

TMEM16A, together with the ezrine–radixin–moesin complex, promotes the formation of a cortical actin cytoskeleton associated with the plasma membrane, a process reported for somatic cells and spermatozoa [19,41]. It is important to note that the presence of a cortical cytoskeleton associated with the plasma membrane and the external acrosomal membrane has been demonstrated [42]. The acrosomal localization of TMEM16A [15] suggests that this Cl^−^ channel could be involved in the establishment of this cortical actin cytoskeleton, unlike CFTR and ClC3 whose subcellular localization has been determined in the equatorial region and the midpiece [16,17,18,19,36]. Additionally, TMEM16A could direct the formation of this cortical actin cytoskeleton through ERK1/2 and RhoA, which are also located in the acrosomal region [10,11]. Therefore, inhibition of TMEM16A would prevent the formation of this cortical actin cytoskeleton and, together with the inhibition of Cl^−^ efflux, would allow the accumulation of Cl^−^ in the cytoplasm, which would also allow water to enter the cytoplasm, with the consequent increase in volume and loss of acrosome structure. Effects were not observed when CFTR and ClC3 were inhibited, as they are not located in the acrosome and although they also block Cl^−^ fluxes and allow Cl^−^ accumulation in the sperm cytoplasm, they did not affect actin polymerization, suggesting that both the inhibition of actin polymerization and the accumulation of Cl^−^ in the cytoplasm are required to alter acrosome structure and increase their volume. The effects of CFTR and ClC3 inhibition would be observed on the flagellum and would be reflected in sperm motility, a fact that has been reported for both CFTR and ClC3.

TMEM16A activity is important for maintaining Cl^−^ anion homeostasis in the cell cytoplasm [43]. In guinea pig sperm, it has been suggested that [Cl^−^]i is important for maintaining HCO_3_^−^ influxes through two families of HCO_3_^−^ transporters, SLC4 and SLC6, where Cl^−^/HCO_3_^−^ exchangers are essential for maintaining a sustained HCO_3_^−^ influx required for capacitation [27,35,44]. This increase in pHi is essential for the influx of Ca^2+^ through CatSper, a Ca^2+^ channel that is activated by the increase in pHi [45]. Both processes are involved in the activation of protein kinase A (PKA), which is responsible for the activation of the signaling cascade that phosphorylates different proteins associated with capacitation and motility [27]. Our results corroborate that during capacitation, intracellular Cl^−^ increases, possibly as a requirement for the influx of HCO_3_^−^ [27,34,44]. The results also show that TMEM16A, CFTR, and ClC3 inhibition increased [Cl^−^]i compared with the capacitated spermatozoa (Figure 5A). The inhibition of these channels may alter intracellular Cl^−^ homeostasis, allowing this anion to flow into the spermatozoa in an uncontrolled manner. One possible consequence of the alteration in Cl^−^ homeostasis is the inhibition of the increase in intracellular pH, leading to the blocking of capacitation and the acrosomal reaction. Therefore, the activities of TMEM16A, CFTR, and ClC3 are relevant to the processes of capacitation and AR (Figure 7A,B).

It has previously been reported that inhibition of TMEM16A and CFTR reduces capacitation, acrosomal reaction, and motility [13,14,15]. Our data corroborate these reports and suggest that this could be an effect of the altered Cl^−^ and pHi homeostasis, which are considered very relevant for these sperm processes [13,14,15,16]. Previous studies have only associated ClC3 with sperm motility [18]. Our results suggest that ClC3 activity is more closely associated with the acrosomal reaction than with capacitation (Figure 7A,B). This indicates a clear differential role for Cl^−^ channels in sperm physiology. Thus, using T16Aihn, it has been shown that TMEM16A is involved in the mechanisms that direct capacitation and AR since the inhibition of TMEM16 during capacitation inhibited this process, and its inhibition after 60 min of capacitation inhibited sAR [13,14]. On the other hand, using CFTRinh, different studies have shown that CFTR is involved in capacitation [16,46,47]. In the case of ClC3, Liu et al. [18] reported that this channel is associated with motility and cell volume regulation in human spermatozoa. Using the same inhibitor (NPPB), we have shown that ClC3 does not participate in capacitation since it follows its normal course despite the spermatozoa being capacitated in the presence of NPPB (Figure 7A); however, the spermatozoa did not undergo AR; therefore, it can be concluded that ClC3 is more involved in the mechanisms that regulate the acrosomal reaction. Further studies are required to define the pathways through which ClC3 may be involved in AR.

Although the inhibition of the three Cl^−^ channels tested had similar effects on sperm physiology, only the inhibition of TMEM16A inhibited actin polymerization, suggesting that TMEM16 activity is more directly associated with the structuring of the actin cytoskeleton. It has been suggested that TMEM16A has a direct relationship with the actin cytoskeleton and that its activity could be modulated by this interaction [19,20]. TMEM16 activates different signaling pathways in several types of cancers, including those associated with RhoA-ROCK and Ras-Raf-MEK-ERK1/2 [22]. It has recently been shown that RhoA activation during capacitation can occur through activation of GEF H1 by ERK1/2 [10]. Based on the results of this study, we postulate that ERK1/2 activity during capacitation depends on TMEM16A and that ERK1/2 activates RhoA through GEF H1, allowing for actin polymerization; thus, inhibition of TMEM16A prevents the activation of ERK1/2 (Figure 6A,B) and, consequently, RhoA and ROCK1, present in guinea pig spermatozoa, are not activated [9] and the actin polymerization required for capacitation does not occur. It is unclear whether there is a direct relationship between TMEM16A and ERK1/2 or whether it is regulated by other elements; thus, further studies are required. We, therefore, propose a new signaling pathway associated with actin polymerization during capacitation that is important for maintaining acrosome structure: TMEM16A-ERK1/2-H1-RhoA-ROCK1. It is important to note that this signaling pathway depends exclusively on TMEM16A and not on other Cl^−^ channels such as CFTR or ClC3.

In conclusion, the present study reveals a new function of the calcium-activated chloride channel (TMEM16A or Ano1) that differs from those associated with intracellular Cl^−^ homeostasis and that affects sperm physiology [13,14]. This new function is required for the activation of the ERK1/2-H1-RhoA-ROCK1-associated signaling pathway; thus, the actin cytoskeleton is remodeled, allowing the acrosomal structure to be maintained prior to the acrosomal reaction.

## 4. Materials and Methods

### 4.1. Reagents

All reagents used in this study were obtained from Sigma-Aldrich (St. Louis, MO, USA) or Bio-Rad Laboratories (Hercules, CA, USA), except where otherwise indicated.

RhoA-GTP was evaluated using a pull-down assay using the RhoA Pull-down Activation Assay Biochem Kit (Cytoskeleton, Denver, CO, USA).

#### 4.1.1. Antibodies

Anti-calreticulin (sc-H104) and anti-pERK1/2 (sc-7383) antibodies were purchased from Santa Cruz Biotechnology Inc. (Santa Cruz, CA, USA). Anti-RhoA (ab68826) was obtained from Abcam (Cambridge, UK), and the anti-ERK1/2 (M5670) antibody was purchased from Sigma-Aldrich (St. Louis, MO, USA). The anti-utrophin 71 antibody was kindly donated by Dr. Dominique Mornet from INSERM U592 (Paris, France).

#### 4.1.2. Inhibitors

TMEM16A inhibitor T16Ainh-A01 (T16Ainh), CFTR inhibitor CFTR-inh (CFTRinh), and ClC3 inhibitor NPPB (5-Nitro-2-(3-phenylpropylamino) benzoic acid) were obtained from Sigma-Aldrich (St. Louis, MO, USA). The ERK1/2 inhibitor FR180204 (sc-203945) was purchased from Santa Cruz Biotechnology Inc. (Santa Cruz, CA, USA). The C3 Rho inhibitor (CT04) was purchased from Cytoskeleton Inc. (Denver, CO, USA).

### 4.2. Experimental Animals

The testicles, epididymis, and vas deferens of male Dunkin Hartley guinea pigs (*Cavia porcellus*) with a mean weight of 800 to 900 g were isolated. Experimental animals were handled in accordance with the Mexican Official Norm for Laboratory Animals (NOM-062-ZOO-1999) and protocols of the Internal Committee for the Care and Use of Laboratory Animals, Cinvestav-IPN (CICUAL No. 321–02), following American Veterinary Medical Association guidelines. Efforts were made to minimize the pain, stress, or distress experienced by the animals.

### 4.3. Capacitation Assay

Spermatozoa were obtained from the vas deferens of guinea pigs and placed in 2 mL of phosphate-buffered saline (PBS, pH 7.4) perfusion. The cells were adjusted to 3.5 × 10^7^ cells/mL and incubated at 37 °C during and until capacitation. TYRODE-HEPES medium (116.7 mM NaCl, 2.8 mM KCl, 1.8 mM CaCl_2_, 0.36 mM NaH_2_PO_4_, 0.49 mM MgCl_2_, 11.9 mM NaHCO_3_, 0.25 mM potassium pyruvate, 0.25 mM sodium pyruvate, 20.0 mM lactic acid, 2.0 mM HEPES, pH 7.6. Osmolarity 310 mOsm/Kg) were used to induce capacitation. For non-capacitive conditions, the cell suspension (3.5 × 10^7^ cells/mL) was incubated at 37 °C in TYRODE-HEPES lacking CaCl_2_ and NaHCO_3_.

### 4.4. Pharmacologic Inhibitors

Different inhibitors used in this work were assessed under capacitive conditions: T16Ainh-A01 (1 µM), CFTR-inh (10 nM), NPPB (25 µM), C3 (1 µg/mL), and FR180204 (10 µM).

### 4.5. Assessment of Sperm Viability

Sperm viability was determined following the method described by Roa-Espitia et al. [38]. Sperm suspensions were incubated in TYRODE-HEPES at 37 °C for 60 min in the absence or presence of T16Ainh, CFTRinh, and NPPB. Once the incubation was complete, a PI solution (1 µg/mL) was added to the spermatozoa sample in a 1:1 ratio and mixed. The mixture was incubated at room temperature for 30 min. The spermatozoa were washed and the numbers of stained and unstained spermatozoa were counted (500 sperm X sample, *n* = 3) under an epifluorescence microscope (Olympus BX500, Tokyo, Japan). Images were registered and analyzed using Nikon Elements 3.1 software.

### 4.6. Assessment of Sperm Head Area

Spermatozoa capacitated under different conditions were fixed with 3% (*v*/*v*) formaldehyde and 0.2% (*v*/*v*) glutaraldehyde in phosphate-buffered saline (PBS). After 60 min, the spermatozoa were collected via centrifugation (600× *g* for 3 min). The pelleted sperm were resuspended in 50 mM NH_4_Cl, incubated for 10 min, and rinsed twice with PBS and then with bi-distilled water. Microscope slides of this suspension were prepared, air-dried overnight at room temperature, and stored at 4 °C. The spermatozoa were stained for 5 min with Coomassie blue G250 (0.22% diluted in an aqueous solution of 5% methane and 10% acetic acid) and then washed three times with distilled water and mounted on glass covers using gelvatol medium for observation. The acrosome area of 100 spermatozoa from each experiment was meticulously assessed using the software Nis Element 3.1. Three experiments (*n* = 3), each with three replicates, were conducted, ensuring the reliability and robustness of our findings.

### 4.7. Immunofluorescence Assays

This procedure was performed as described by Roa-Espitia et al. [36]. Spermatozoa capacitated under different conditions were fixed with 3% (*v*/*v)* formaldehyde and 0.2% (*v*/*v*) glutaraldehyde in phosphate-buffered saline (PBS). After 1 h, the sperm were collected via centrifugation (600× *g* for 3 min). The pelleted sperm were incubated in 50 mM NH_4_Cl for 10 min and rinsed twice with PBS and then with bi-distilled water. Microscope slides of this suspension were prepared, air-dried overnight at room temperature, and stored at 4 °C. Sperm cells were permeabilized in acetone for 7 min at −20 °C and washed with PBS. Anti-calreticulin antibody (1:50) was diluted in PBS with 1% BSA (blocking solution) and incubated on the slides overnight at room temperature. After extensive washes with PBS, the cells were incubated for 2 h at 37 °C with the appropriate TRITC (ex/em = 557/576 nm) conjugated secondary antibody diluted in blocking solution. The spermatozoa on the slides were extensively washed with PBS. The samples were then mounted on glass coverslips using gelvatol, sealed adequately, and stored at −20 °C until observed. The stained cells were imaged under a confocal laser scanning microscope (Leica TCS SP8, Vienna, Austria) and analyzed using the imaging software LAS AF Lite (Ver. 2.6.3).

### 4.8. Intracellular Measurement of Cl^−^ in the Sperm Population

The intracellular concentration of Cl^−^ ([Cl^−^]i) was evaluated following the method described by Cordero-Martínez et al. [48]. Spermatozoa were loaded with 10 μM Cl^−^-sensitive fluorescent dye N-(ethoxycarbonylmethyl)-6-methoxyquinolinium bromide (MQAE) and incubated under conditions that do not support capacitation (Cl^−^-free medium: 116.7 mM sodium gluconate, 2.8 mM potassium gluconate, 0.36 mM KH_2_PO_4_, 0.49 mM MgSO_4_, 0.25 mM pyruvic acid, 20 mM lactic acid, and 20 mM HEPES) for 30 min at 37 °C and pH 7.2. Excess MQAE was removed by diluting sperm 10-fold with Cl^−^-free medium and centrifuging for 3 min at 600× *g*. Sperm pellets were resuspended in aliquots of 35 × 10^6^ sperm/mL with a relevant medium. To estimate [Cl^−^]i, spermatozoa were loaded with MQAE (ex/em = 355/460 nm) and incubated for 30 min in mediums containing different concentrations of Cl^−^ (0–100 mM). Excess MQAE was removed and fluorescence emission intensity data for the sperm suspensions were recorded for 3 min. The fluorescence measurements were carried out in a Synergy 2 Multi-Function Microplate Reader (Bio-Tek Instruments, Winooski, VT, USA); samples were illuminated at 350 nm, and emission was recorded at 450 nm. The influence of T16Ainh, CFTRinh, and NPPB on [Cl^−^]i was determined using spermatozoa loaded with MQAE. Spermatozoa were resuspended in TYRODE-HEPES solution and incubated in the presence or absence of T16Ainh, CFTRinh, or NPPB. Spermatozoa were incubated for 60 min, and their basal fluorescence was recorded. Two control experiments were performed: (1) the fluorescence of water was measured, and (2) MQAE fluorescence without cells was recorded, followed by the addition of T16Ainh, CFTRinh, and NPPB. This procedure was conducted to evaluate the difference in the fluorescence of T16Ainh, CFTRinh, and NPPB compared with that of water only. The readings were interpolated in a Cl^−^ calibration curve.

### 4.9. Intracellular pH Measurement in the Sperm Population

The intracellular pH (pHi) was determined following the methodology described by Cordero-Martínez et al. [48]. Sperm samples were incubated for 30 min at 37 °C with the pHi fluorescent indicator BCECF-AM (ex/em = 500/531 nm). The samples were then washed via centrifugation (600× *g* for 6 min) to remove excess fluorescent indicator and resuspended in fresh Tyrode media. Then, spermatozoa were transferred into a 96-well plate. After 60 min of incubation under capacitating conditions, absorbance was measured using Synergy 2 Multi-Function Microplate Reader (Bio-Tek Instruments, Winooski, VT, USA) at wavelengths of 490 nm of excitation and 535 nm of emission. All data obtained were interpolated in an activity calibration curve generated using the same sperm concentration (8–10 × 10^6^ cells/mL) and incubated with modified TYRODE media with different pH values (7.0, 7.2, 7.4, 7.6, 7.8, and 8.0). Before recording fluorescence, permeabilization of the spermatozoa was performed with 0.1% Triton. The influences of T16Ainh, CFTRinh, and NPPB on pHi were determined using spermatozoa loaded with BCECF-AM. The spermatozoa were resuspended in TYRODE-HEPES solution and incubated in the presence or absence of T16Ainh, CFTRinh, or NPPB for 60 min; their basal fluorescence was then recorded.

### 4.10. Assessment of F-Actin

Spermatozoa were fixed in 1.5% formaldehyde in PBS for 1 h at room temperature and collected via centrifugation at 600× *g* for 4 min. The pelleted spermatozoa were immediately resuspended in 50 mM NH_4_Cl in PBS, incubated for 10 min, and washed twice by resuspension/centrifugation in PBS. Sperm smears were prepared and stained. Sperm on the smears were permeabilized at 20 °C for 7 min using acetone and washed with PBS. F-actin was observed using phalloidin-FITC (ex/em = 495/519 nm): the spermatozoa were incubated with 30 μM phalloidin-FITC at room temperature for 60 min. The smears were then exhaustively washed with PBS and mounted on glass covers using gelvatol for observation. Images were acquired using an Olympus BX50 (Tokio, Japan) photomicroscope equipped with phase contrast and epifluorescence or a Leica confocal microscope. Fluorescence was analyzed using Nikon Nis 3.0 software. Fluorescence quantification was only carried out on the sperm head.

### 4.11. Rhotekin–Rho Binding Domain Pull-Down Assay

This procedure was performed as described by Ramírez-Ramírez et al. [7]. To evaluate the activation of RhoA, RhoA-GTP was recovered using RhoA Rhotekin-RBD Pull-down Activation Assay Biochem Kit (No. BK036-S; Cytoskeleton). Protein extracts (300 μg) of sperm capacitated under different conditions were incubated with 50 μg of agarose-conjugated Rhotekin–Rho binding domain (RBD) at 4 °C for 1 h. Proteins not bound to the Rhotekin–RBD were recuperated via centrifugation at 5000× *g* at 4 °C for 3 min after two washes. Rhotekin–RhoA-GTP precipitates were diluted in Laemmli sample buffer and then boiled to release active RhoA, which was separated using SDS-PAGE. Next, proteins were transferred to nitrocellulose membranes and analyzed by immunoblotting using the relevant antibodies.

### 4.12. Immunoblotting

Immunoblotting was performed as described by Roa-Espitia et al. [36]. Cells (350 × 106) were suspended in lysis buffer (50 mM Tris–HCl at pH 7.4, 1 mM EGTA, 1 mM PMSF, complete protease inhibitor cocktail, 1 mg/mL aprotinin, 10 mM sodium orthovanadate, 25 mM sodium fluoride, and 1% Triton X-100) as previously described. The samples were centrifuged at 5000× *g* for 5 min at 4 °C, and the protein concentrations of the supernatant fractions were determined as described previously. The samples were then boiled for 5 min in sample buffer, resolved on 7% or 10% SDS-PAGE gels, and transferred onto PVF or nitrocellulose membranes as Roa-Espitia et al. [36]. The membranes were blocked using Tris-buffered saline (TBS) containing 0.1% Tween-20 and 5% fat-free dry milk. The membranes were incubated overnight at 4 °C with the respective antibodies (anti-Rho A (1:1000), anti-ERK1/2 (1:1000), or anti-pERK1/2 (1:1000)). After incubation, the membranes were washed five times (7 min each time) and then incubated with the appropriate HRP-labeled secondary antibody (1:10,000). Immunoreactive proteins were detected using chemiluminescence with a Millipore-ECL Western blot detection kit (Merk, KGaA, Darmstadt, Germany).

### 4.13. Evaluation of Capacitation Status by Staining with Chlortetracycline (CTC)

The CTC assay was performed following the method described by Roa-Espitia et al. [36]. Briefly, the staining solution was prepared by dissolving 250 μM CTC-HCl in TN buffer (20 mM Tris, 130 mM NaCl, and 5 mM cysteine at pH 7.8); fresh CTC stock was prepared daily. Sperm suspensions (20 μL) were added to the same volume of pre-warmed (37 °C) CTC stock solution and incubated for 10 s; 3.5 μL of 12.5% glutaraldehyde in 1.25 M Tris buffer (pH 7.5) was added and immediately followed by gentle mixing. Fixed samples were kept in a dark box. Slides were prepared 1–4 h after fixation and examined under a fluorescence microscope (excitation at 330–380 nm, emission at 420 nm). A total of 250 spermatozoa were counted to assess previously recognized CTC staining patterns [49]: F = non capacitated, B = capacitated with an intact acrosome, and AR = undergone spontaneous acrosome reaction.

### 4.14. Statistical Analysis

Each experiment was conducted at least in triplicate. To determine the normality of the data, the Shapiro–Wilk test was performed. Then the statistical significance was analyzed using one-way ANOVA after Tukey’s or *t*-test to compare multiple groups or two groups, respectively. SigmaPlot version 11.0 software was used for the analysis. All data are presented as mean ± S.E.; *p* < 0.05 was considered statistically significant.

## Figures and Tables

**Figure 1 ijms-26-03750-f001:**
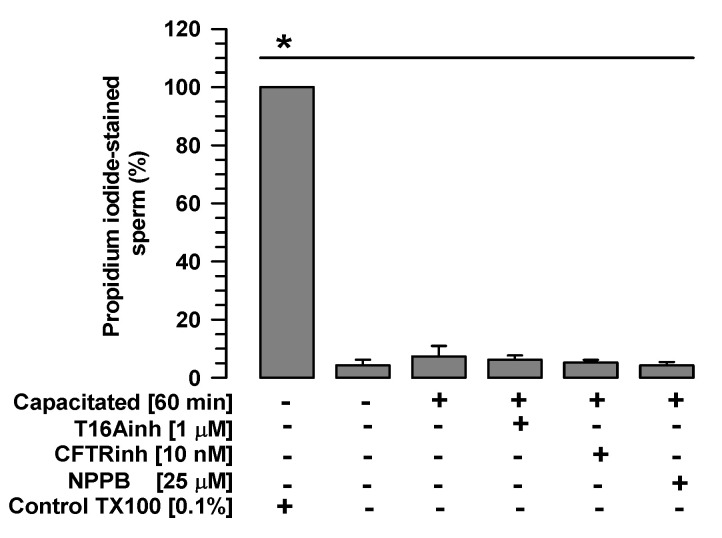
Sperm viability is not affected by TMEM16A, CFTR, or ClC3 inhibitors. Guinea pig spermatozoa were capacitated for 60 min in the presence of one of the following Cl^−^ channel inhibitors: T16Ainh (1 µM), CFTRinh (10 nM), or NPPB (25 µM). Viability was assessed using propidium iodide, and the number of stained and unstained spermatozoa was counted (500 cells per sample). Images were recorded and analyzed using Niko Element 3.1 software. Results are expressed as mean ± S.E. (*n* = 3), * *p* < 0.05.

**Figure 2 ijms-26-03750-f002:**
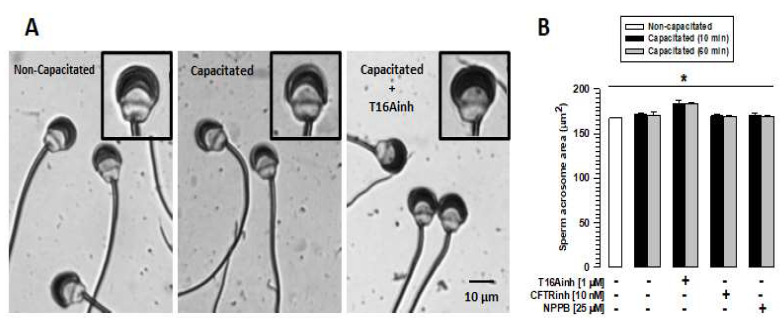
TMEM16A inhibition affects the volume of the acrosomal region. Guinea pig spermatozoa were capacitated for 60 min in the presence of T16Ainh, CFTRinh, or NPPB inhibitors. Spermatozoa were then fixed and stained with Coomassie blue G250. The acrosomal area of 100 spermatozoa was carefully assessed using Nis Element 3.1 software. (**A**) Images of non-capacitated and capacitated spermatozoa in the absence and presence of T16Ainh. The inset shows the acrosomal changes experienced by spermatozoa capacitated in the presence of T16Ainh (right panel) compared to non-capacitated (left panel) and spermatozoa capacitated in the absence of T16Ainh (middle panel). (**B**) Quantification of changes in the acrosome area of capacitated spermatozoa in the absence and presence of the different inhibitors. Results are expressed as means ± S.E. (*n* = 3), * *p* < 0.05.

**Figure 3 ijms-26-03750-f003:**
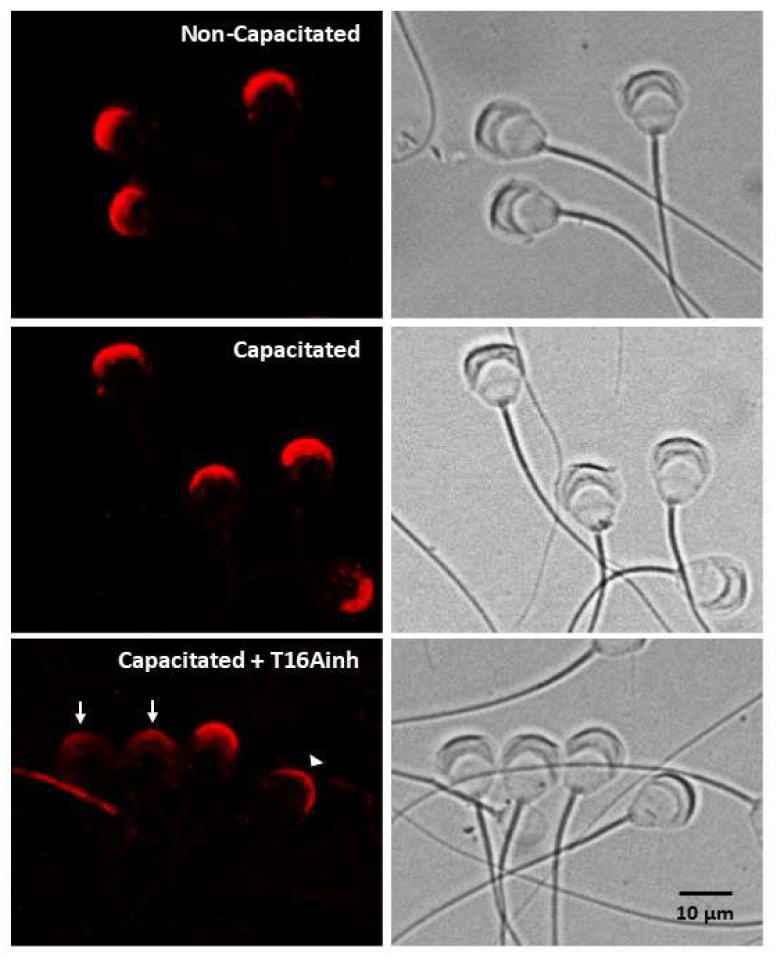
TMEM16A inhibition disrupts acrosomal structure. A specific antibody for the acrosomal protein calreticulin was used to define the changes in the acrosome produced when spermatozoa were capacitated for 60 min in the absence or presence of the inhibitor T16Ainh (1 µM). The localization of calreticulin is shown in the left panels. The top panel shows non-capacitated spermatozoa, the middle panel shows capacitated spermatozoa, and the bottom panel shows spermatozoa capacitated in the presence of T16Ainh. The right panels show bright field images of spermatozoa. Images are representative of three different experiments. Arrow: pattern with dim fluorescence dispersed throughout the acrosome. Arrowhead: pattern with fluorescence concentrated in the apical region of the acrosome.

**Figure 4 ijms-26-03750-f004:**
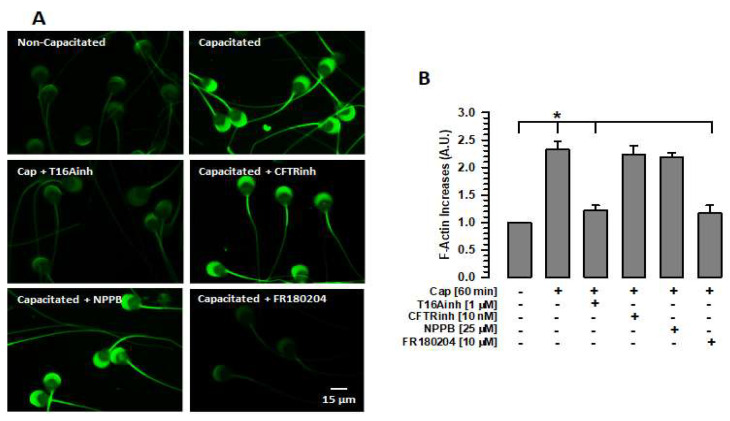
Inhibition of TMEM16A or ERK1/2 prevents actin polymerization during capacitation. To define the changes in F-actin levels produced by inhibition of Cl^−^ channels or ERK1/2, spermatozoa were capacitated in the absence or presence of one of the following inhibitors: T16Ainh, CFTRinh, NPPB, or FR180204 (ERK1/2). After 60 min of incubation, spermatozoa were fixed, and F-actin was visualized by staining with phalloidin-FITC. (**A**) Localization of F-actin using phalloidin-FITC. Images represent three independent experiments. (**B**) Quantification of fluorescence emitted by phalloidin-FITC-stained spermatozoa. Cap: capacitated. Fluorescence levels were assessed using Nis Element 3.1 software. The fluorescence values were normalized with respect to values of non-capacitated spermatozoa. Results are expressed as mean ± S.E. (*n* = 3), * *p* < 0.05.

**Figure 5 ijms-26-03750-f005:**
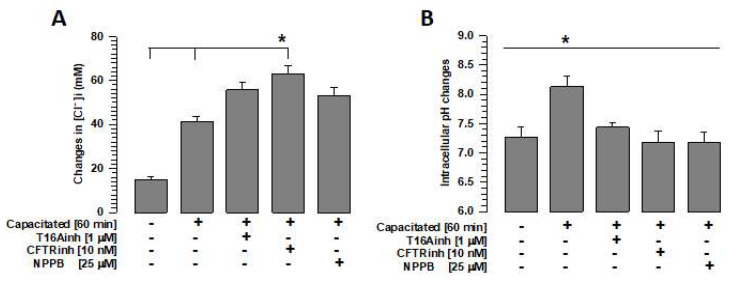
Inhibition of chloride ion channels alters intracellular Cl^−^ and pH homeostasis. To define the effects of TMEM16A, CFTR, and ClC3 channel inhibition on intracellular Cl^−^ and pH homeostasis, spermatozoa were capacitated for 60 min in the absence or presence of the inhibitors T16Ainh, CFTRinh, or NPPB, and intracellular Cl^−^ and intracellular pH were assessed using specific probes. (**A**) Quantification of changes in intracellular Cl^−^ concentrations using MQAE. Results are expressed as mean ± S.E. (*n* = 5), * *p* < 0.05. (**B**) Quantification of intracellular pH changes using BCECF-AM. Results are expressed as mean ± S.E. (*n* = 5), * *p* < 0.05.

**Figure 6 ijms-26-03750-f006:**
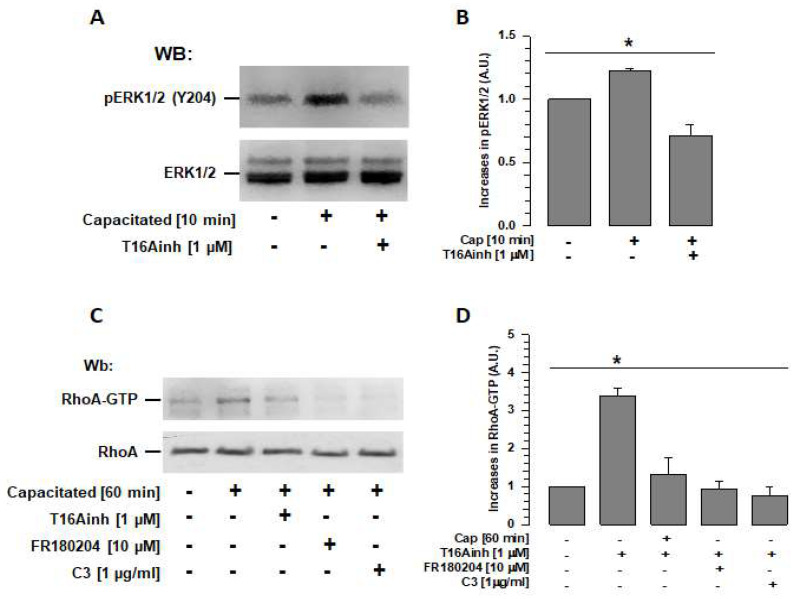
TMEM16A regulates RhoA activity through ERK1/2 during capacitation. (**A**) Total extracts of spermatozoa capacitated for 10 min in the absence or presence of the TMEM16A inhibitor (T16Ainh) were used to determine its effect on ERK1/2 activity. Ten minutes of capacitation is the time when ERK1/2 reaches their maximum activity. Specific antibodies were used to detect ERK1/2 (pERK) phosphorylated at Y204 (upper panel) and total ERK1/2 in each sample (lower panel). WBs represent three independent experiments. (**B**) WBs of ERK1/2 and pERK1/2 were analyzed using densitometry. Results were first normalized using the ratio N/N0, where N is the amount of pERK1/2 and N0 is the total amount of ERK1/2. The results obtained were then normalized with respect to the results of non-capacitated sperm. Results are expressed as mean ± S.E. (*n* = 3), * *p* < 0.05. (**C**) Total extracts of spermatozoa capacitated for 60 min in the absence or presence of the ERK1/2-specific inhibitor FR180204 were used to determine its effect on RhoA activity. RhoA-GTP was isolated via pull-down using a Rhotekin column and analyzed using WB and a RhoA-specific antibody (upper panel). The lower panel shows the total amount of RhoA in each sample. WBs represent three independent experiments. (**D**) WBs for RhoA-GTP and RhoA were analyzed using densitometry. Cap: capacitated. Results were first normalized as the ratio N/N0, where N is the amount of RhoA-GTP and N0 is the total amount of RhoA, and then normalized to the results of non-capacitated spermatozoa. Results are expressed as mean ± S.E. (*n* = 3), * *p* < 0.05. Cap: capacitated.

**Figure 7 ijms-26-03750-f007:**
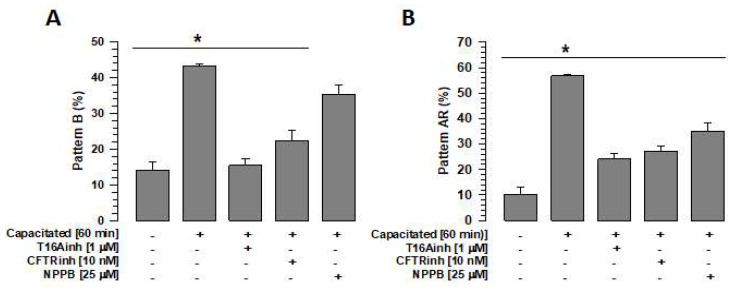
Inhibition of Cl^−^ channels, TMEM16A, CFTR, and ClC3 affects capacitation and the acrosome reaction. To define the effects of inhibition of TMEM16A, CFTR, and ClC3 channels on capacitation and the spontaneous acrosome reaction, spermatozoa were capacitated for 60 min in the absence or presence of the inhibitors T16Ainh, CFTRinh, or NPPB, and the B and AR patterns were assessed by CTC staining. (**A**) Effects of the inhibitors T16Ainh, CFTRinh, and NPPB on the B pattern. Results are expressed as mean ± S.E. (*n* = 3), * *p* < 0.05. (**B**) Effects of the inhibitors T16Ainh, CFTRinh, and NPPB on the AR pattern. Results are expressed as mean ± S.E. (*n* = 3), * *p* < 0.05.

## Data Availability

Data are contained within the article and Appendix A.

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
