# Peer review of "TMEM16A Maintains Acrosomal Integrity Through ERK1/2, RhoA, and Actin Cytoskeleton During Capacitation"

_ijms, 2025, doi:10.3390/ijms26083750_

Round 1
Reviewer 1 Report
Comments and Suggestions for Authors
The manuscript by Roa et al. addresses some aspects of Chloride channels' participation in mammalian sperm physiology. In general, this work is a follow-up to the author's previous work on Cl- channels and actin remodeling in mammalian sperm. The authors indicate that TMEM16A, but not CFTR and CLIC3, is relevant for acrosome volume regulation and actin cytoskeleton dynamics.
Although some of the experiments are executed properly, some gaps need to be addressed, and in particular the physiological significance of the findings is not clear and even puzzling. I have some major concerns with this manuscript.
The results in Fig. 1 indicate that only TMEM16A is involved in acrosome volume regulation. Some questions arise from this experiment:
- In other mammalian species, acrosome swelling has been proposed as a requirement for acrosome reaction, why do the authors think there are no differences and almost no variability observed in the volume between NC and CAP cells?
- In Figure 7, the authors show that the AR pattern in CAP cells is around 60%, the area (volume) measurements were performed in intact sperm alone (the remaining 40%), reacted alone or no distinction was considered.
- I consider that control conditions for volume changes can potentiate the impact of the results, similar to the results of Fig. 4 (using FR180204), and Fig 6C and D using FR180204 and C3 compounds.
The next central result is shown in Fig. 4, the authors indicate that only TMEM16A inhibition affects actin cytoskeleton remodeling. However, the description of the results in Fig. 4 is not clear enough.
- Was the fluorescence quantification performed in the acrosome region only? or in the whole cell? If the latter is true, those results can be masking the effect at the acrosome level.
- The authors discuss the relationship of TMEM16A function with the cytoskeleton remodeling process by connecting the channel's activity with signaling proteins (ERK pathway), however, it is still not clear how this can explain acrosome volume changes. Would the treatment with FR180204 induce a similar behavior as TMEM16A inhibition? Is TMEM16A localized in the acrosomal membrane? If not, how the signals are transduced from the plasma membrane to the acrosome?
Fig. 5 shows that all the tested inhibitors produce (at different extents) the same physiological behavior, meaning an increase in the intracellular Cl- levels and a decrease in the pHi. From lines 362-364, the authors discuss that TMEM16A, CFTR, and CLIC3 inhibition paradoxically causes "uncontrolled chloride flow" leading to intracellular Cl- increase. I suggest the authors discuss this more openly, citing the relevant studies along with the differences in anion conductances for each tested channel, as well as the role of intracellular localization of each protein. A schematic model would be helpful.
Minor points:
- In line 104, the sentence "TMEM16A, CFTR, and NPPB inhibitors were assessed" seems to indicate the name of the inhibitors and the name of the proteins inhibited by each.
- When referring to signal intensity "dim" should be used instead of 'pale'
- In Fig. 3 (bottom) some arrows and arrowheads are shown but it is not indicated the meaning of this.
- The main text, says that phalloidin-FITC was employed, however, in the figure legend and the methods sections is indicated that phalloidin-TRIC was used. Clarify.
- In the main text, nothing is commented about the results of FR180204
- In line 205 some numerical data is presented but there is no explanation about the units of this.
- Lines 214-215 claim "These results suggest that Cl- fluxes are essential for maintaining intracellular Ca2+ concentration..." However, no calcium measurements were done.
- Figures 4A and 6A indicate that the concentration of T16Inh was 10 uM, however, other Figures say is 1 uM. Clarify if different concentrations were employed and why.
- Fig 6B and D indicate that the units are A.U, however, this should indicate instead that it is the N/N0 ratio.
- Line 327 says TMEM16F, not A.
- Is the age of the animals employed considered, or only the weight?
- In sections 4.5, 4.7, and 4.10 the imaging conditions such as the ex/em information and objective magnification need to be included.
- The final Osmolarity of the media needs to be included.
- Line 465 and 536 "NH4Cl" lacks the subindex.
- The statistical analysis is not clear, no normality tests were performed to decide for parametric or non-parametric tests. Absolute p values are preferred to <0.05. The indications "* vs ** P<0.01" are not clear enough.
Language should be improved, many typos are present and seems that a direct Spanish-to-English translation was used (i.e. Fig. 6A says "capacitados" instead of capacitated, "membrane plasma' instead of plasma membrane).
Author Response
México City on April 5, 2025
Prof. Dr. Maurizio Battino
Editor in Chief
International Journal of Molecular Science
Regarding the manuscript IJMS-35297 entitled " TMEM16A Maintains Acrosomal Integrity through ERK1/2, RhoA, and Actin Cytoskeleton During Capacitation " by Ana L. Roa-Espitia, Tania Reyes-Miguel, Monica L. Salgado-Lucio1, Joaquín Cordero-Martínez, Dennis Tafoya-Domínguez, Enrique O. Hernández-González. We have carefully considered all comments and suggestions and provided an appropriate response. We also modified the manuscript where necessary. The manuscript has been improved and can now be ready for publication in the International Journal of Molecular Science.
.
We thank the reviewers for their positive and constructive evaluation of our work.
All changes made to the manuscript are marked. We look forward to your answer.
Sincerely.
Enrique O Hernández-González PhD.
Senior Researcher
Cinvestav, Unidad Zacatenco.
Av. Instituto Politécnico Nacional 2508
San Pedro Zacatenco, Ciudad de México, CP 0763. México.
Phone: 52-5557473800, Ext. 3352
Email: enrique.hernandez@cinvestav.mx
Comments and Suggestions for Authors [jms-3529707]
The manuscript by Roa et al. addresses some aspects of Chloride channels' participation in mammalian sperm physiology. In general, this work is a follow-up to the author's previous work on Cl- channels and actin remodeling in mammalian sperm. The authors indicate that TMEM16A, but not CFTR and CLIC3, is relevant for acrosome volume regulation and actin cytoskeleton dynamics.
Although some of the experiments are executed properly, some gaps need to be addressed, and in particular the physiological significance of the findings is not clear and even puzzling. I have some major concerns with this manuscript.
First, we thank the reviewer for his insightful comments and suggestions, which have improved this manuscript. We have followed his suggestions and comments to the fullest.
The results in Fig. 1 indicate that only TMEM16A is involved in acrosome volume regulation. Some questions arise from this experiment:
Comment 1. In other mammalian species, acrosome swelling has been proposed as a requirement for acrosome reaction, why do the authors think there are no differences and almost no variability observed in the volume between NC and CAP cells?
Answer. Zanetti and Mayorga (DOI 10.1095/biolreprod.109.076166) have shown that the acrosomal reaction requires acrosome swelling, as this process allows the plasma membrane and the external acrosomal membrane to merge. This process only occurs after sperm have undergone two hours of capacitation and the acrosomal reaction is induced with A23187. It is important to consider that during capacitation, an actin cytoskeleton is structured between the plasma and external acrosomal membranes, which prevents the membranes from docking and the sperm from undergoing spontaneous acrosomal reaction (Breitbart et al., 2005 [doi: 10.1530/rep.1.00269.]; Ramírez-Ramírez et al., 2020 [DOI: 10.1002/jcb.29521]). This cytoskeleton is destructured when inducers of the acrosomal reaction (A23187, Progesterone or ZP3) trigger signaling pathways that elevate the intracellular concentration of calcium, allowing the activation of enzymes such as gelsolin and scinderin (Pelletier et al., 199 [doi: 10.1095/biolreprod60.5.1128.]; Finkelstein et al., 2010 [doi: 10.1074/jbc.M110.170951. ]) that fragment the actin cytoskeleton and allow the swelling of the acrosome with the consequent coupling of the plasma and external acrosomal membranes. Therefore, there should not be a significant difference in volume between non-capacitated and capacitated spermatozoa, as long as the actin cytoskeleton is normally structured.
Comment 2. In Figure 7, the authors show that the AR pattern in CAP cells is around 60%, the area (volume) measurements were performed in intact sperm alone (the remaining 40%), reacted alone or no distinction was considered.
Answer: For the assessment of acrosomal volume, only sperm that had not undergone acrosomal reaction, i.e., those that retained their acrosome, were considered. To avoid confusion, we have now included the results based on the change in acrosomal volume at 10 minutes of incubation. This time was selected because an increase in volume can be observed after 10 minutes of capacitation (Fig. 2B), and when approximately 90% of sperm retained their acrosome (Supplemental Figure 1).
Comment 3. I consider that control conditions for volume changes can potentiate the impact of the results, similar to the results of Fig. 4 (using FR180204), and Fig 6C and D using FR180204 and C3 compounds.
Answer: While inhibition of ERK1/2 by FR180204 or of RhoA by C3 completely or partially blocks actin polymerization, they appear not to affect Cl- or other ion fluxes. Therefore, it could be suggested that there would be no appreciable effect on sperm volume. However, this would need to be studied.
The next central result is shown in Fig. 4, the authors indicate that only TMEM16A inhibition affects actin cytoskeleton remodeling. However, the description of the results in Fig. 4 is not clear enough.
Comment 4. Was the fluorescence quantification performed in the acrosome region only? or in the whole cell? If the latter is true, those results can be masking the effect at the acrosome level.
Answer: Fluorescence quantification was only performed on the sperm heads. It has been clarified in section 4.7 of the methods (lines 572-573).
Comment 5. The authors discuss the relationship of TMEM16A function with the cytoskeleton remodeling process by connecting the channel's activity with signaling proteins (ERK pathway), however, it is still not clear how this can explain acrosome volume changes. Would the treatment with FR180204 induce a similar behavior as TMEM16A inhibition? Is TMEM16A localized in the acrosomal membrane? If not, how the signals are transduced from the plasma membrane to the acrosome?
Answer. We believe that TMEM16A inhibition has several important effects: the first and well-known one is the relationship between TMEM16A activity and ERK1/2 activity through the Ras/Raf/MEK signaling pathway (Wan et al., 2017. DOI 10.1186/s12943-017-0720-x), a signaling pathway present in mammalian sperm (Lamirande and Gagnon 2002 [doi: 10.1093/molher/8.2.124.], líneas 74-91). Second, our group recently reported the relationship of ERK1/2 with the activation of the GEF H1, which is related to the activation of RhoA (Salgado-Lucio et al., 20220 [doi:10.1242/jcs.239186]). Finally, TMEM16A activity participates in [Cl-]i homeostasis, such that inhibiting this calcium-dependent Cl- channel prevents Cl- effluxes (Schreiber et al., 20204. https://doi.org/10.1016/j.ceca.2024.102885), then Cl- is concentrated in the cytoplasm allowing the entry of water and, together with the absence of the actin cytoskeleton, acrosome volume is altered. On the other hand, although the inhibition of ERK1/2 by FR180204 prevents actin polymerization during capacitation, it does not prevent Cl- fluxes, so if there were an increase in cell volume, it might not be appreciable. The above has been explained in the discussion (lines 334-377).
As to whether the actin cytoskeleton is associated with the acrosomal membrane, the answer is yes, it is discussed in lines 358-375.
Comment 6. Fig. 5 shows that all the tested inhibitors produce (at different extents) the same physiological behavior, meaning an increase in the intracellular Cl- levels and a decrease in the pHi. From lines 362-364, the authors discuss that TMEM16A, CFTR, and CIC3 inhibition paradoxically causes "uncontrolled chloride flow" leading to intracellular Cl- increase. I suggest the authors discuss this more openly, citing the relevant studies along with the differences in anion conductances for each tested channel, as well as the role of intracellular localization of each protein. A schematic model would be helpful.
Answer: We have taken the reviewer's suggestion into account and analyzed the relationship between the subcellular localization of TMEM16A, CFTR, and ClC3 with the alteration of acrosome structure and volume (lines 358–377).
The relationship between the conductance of TMEM16A, CFTR, and ClC3 and Cl⁻ accumulation in the sperm cytoplasm was not included, as it requires a broad discussion that would involve not only these three channels but also other ion cotransporters such as NKCC, or antiporters as Cl⁻/HCO⁻, among others. This would result in a very extensive discussion.
Minor points:
Comment 7. In line 104, the sentence "TMEM16A, CFTR, and NPPB inhibitors were assessed" seems to indicate the name of the inhibitors and the name of the proteins inhibited by each.
Answer: The sentence "TMEM16A, CFTR, and NPPB inhibitors were assessed" was corrected to "TMEM16A, CFTR, and ClC3 inhibitors were assessed" (line 111).
Comment 8. When referring to signal intensity "dim" should be used instead of 'pale'
Answer: “pale” was changed to “dim” (lines 157 and 182).
Comment 9. In Fig. 3 (bottom) some arrows and arrowheads are shown but it is not indicated the meaning of this.
Answer: The meaning of the arrows and arrow heads are now indicated in Fig. 3 (lines 170-172).
Comment 10. The main text, says that phalloidin-FITC was employed, however, in the figure legend and the methods sections is indicated that phalloidin-TRIC was used. Clarify.
Answer: This error was corrected "phalloidin-TRIC" was changed to "phalloidin-FITC" (lines 198-200).
Comment 11. In the main text, nothing is commented about the results of FR180204
Answer: The results about the effect of FR180204 are found in lines 262-268.
Comment 12. In line 205 some numerical data is presented but there is no explanation about the units of this.
Answer: The units of this data were included (lines 214-215).
Comment 13. Lines 214-215 claim "These results suggest that Cl- fluxes are essential for maintaining intracellular Ca2+ concentration..." However, no calcium measurements were done.
Answer: This claim was changed by “These results suggest that Cl- fluxes are essential for maintaining the pHi homeostasis” (lines 223-224).
Comment 14. Figures 4A and 6A indicate that the concentration of T16Inh was 10 uM, however, other Figures say is 1 uM. Clarify if different concentrations were employed and why.
Answer: This error has been corrected in Fig. 4A and 6A. The concentration used in this study was 1 µM, which is the optimal concentration indicated by the supplier for inhibiting TMEM16A. Higher concentrations may produce undesirable side effects.
Comment 15. Fig 6B and D indicate that the units are A.U, however, this should indicate instead that it is the N/N0 ratio.
Answer: First, the WB densitometric data were analyzed using the N/N0 ratio. The data obtained from this ratio were then normalized with respect to the data from the non-capacitated spermatozoa; therefore, A.U. was used.
Comment 16. Line 327 says TMEM16F, not A.
Answer: “TMEM16F” was changed to “TMEM16A” (line 336).
Comment 17. Is the age of the animals employed considered, or only the weight?
Answer: In our experience, although guinea pigs reach sexual maturity between two and three months of age, we have found that, starting at five months of age, when they reach a weight of 800 to 900 grams, sperm production and quality are optimal for our experiments. For this reason, we use guinea pigs weighing between 800 and 900 grams. This also allows us to use the smallest number of organisms.
Comment 18. In sections 4.5, 4.7, and 4.10 the imaging conditions such as the ex/em information and objective magnification need to be included.
Answer: In sections 4.7 (lines 514-515), 4.8 (line 531), 4.9 (line 549), and 4.10 (lines 570), information about the ex/em of the fluorochromes used has been added.
Comment 19. The final Osmolarity of the media needs to be included.
Answer: The osmolarity of the Tyrode-HEPES medium was included and is similar to that reported by Rogers and Yanagimachi, 1975 (https://doi.org/10.1095/biolreprod13.5.568). Line 472.
Comment 20. Line 465 and 536 "NH4Cl" lacks the subindex.
Answer: This error was corrected, lines 494, 509, and 567.
Comment 21. The statistical analysis is not clear, no normality tests were performed to decide for parametric or non-parametric tests. Absolute p values are preferred to <0.05. The indications "* vs ** P<0.01" are not clear enough.
Answer: Although not indicated in section 4.14, the Shapiro-Wilk test was performed to determine data normality, followed by a one-way ANOVA after Tukey's analysis to compare multiple groups and a Student t-test for two groups. Only absolute values of P <0.05 were considered (lines 622-625).
Comments on the Quality of English Language
Comment 22. Language should be improved, many typos are present and seems that a direct Spanish-to-English translation was used (i.e. Fig. 6A says "capacitados" instead of capacitated, "membrane plasma' instead of plasma membrane).
Answer: The typos found have been corrected:
“membrane plasma” has been changed to “plasma membrane” (line 77).
Figure 6A has been corrected, “capacitados” has been changed to “capacitated.”
The English language has been corrected by the MDPI Editing Service, certificate No. 90548.

Reviewer 2 Report
Comments and Suggestions for Authors
The manuscript entitled "TMEM16A Maintains Acrosomal Integrity through ERK1/2, RhoA, and Actin Cytoskeleton During Capacitation" provides new insights into the role of the Ca²⁺-activated Cl⁻ channel TMEM16A in sperm capacitation, particularly in regulating actin polymerization and maintaining acrosomal integrity. The mechanistic exploration of TMEM16A via ERK1/2 and RhoA signaling is novel and of physiological relevance. The experimental data are generally convincing and well-presented, and the manuscript contributes meaningfully to the understanding of ion channel signaling during mammalian fertilization. However, several aspects of the experimental design and mechanistic interpretation would benefit from additional clarification or further validation to reinforce the study's overall impact and biological depth.
- While the study used T16Ainh-A01 as a TMEM16A inhibitor, this compound may have off-target effects. Have the authors considered validating the findings using a second TMEM16A-specific inhibitor or siRNA knockdown approaches to confirm that the observed phenotypes (e.g., disrupted acrosomal structure and reduced actin polymerization) are truly TMEM16A-dependent?
- The authors show that TMEM16A, CFTR, and ClC3 all contribute to intracellular Cl⁻ and pH homeostasis, but only TMEM16A regulates actin polymerization. Could the authors discuss why CFTR and ClC3 do not affect actin cytoskeleton remodeling, despite altering ionic homeostasis? This might reflect distinct spatial or temporal activation of these channels during capacitation.
- In addition, the manuscript would greatly benefit from the inclusion of a schematic summary or workflow diagram that visually illustrates the proposed TMEM16A–ERK1/2–RhoA–actin axis during capacitation. This would help readers better grasp the experimental logic and mechanistic model at a glance. The authors may refer to recent visual schematics used in MDPI journals (e.g., PMID: 35328243 , 30669548) for inspiration.
Author Response
Comments and Suggestions for Authors [jms-3529707]
The manuscript by Roa et al. addresses some aspects of Chloride channels' participation in mammalian sperm physiology. In general, this work is a follow-up to the author's previous work on Cl- channels and actin remodeling in mammalian sperm. The authors indicate that TMEM16A, but not CFTR and CLIC3, is relevant for acrosome volume regulation and actin cytoskeleton dynamics.
Although some of the experiments are executed properly, some gaps need to be addressed, and in particular the physiological significance of the findings is not clear and even puzzling. I have some major concerns with this manuscript.
First, we thank the reviewer for his insightful comments and suggestions, which have improved this manuscript. We have followed his suggestions and comments to the fullest.
The results in Fig. 1 indicate that only TMEM16A is involved in acrosome volume regulation. Some questions arise from this experiment:
Comment 1. In other mammalian species, acrosome swelling has been proposed as a requirement for acrosome reaction, why do the authors think there are no differences and almost no variability observed in the volume between NC and CAP cells?
Answer. Zanetti and Mayorga (DOI 10.1095/biolreprod.109.076166) have shown that the acrosomal reaction requires acrosome swelling, as this process allows the plasma membrane and the external acrosomal membrane to merge. This process only occurs after sperm have undergone two hours of capacitation and the acrosomal reaction is induced with A23187. It is important to consider that during capacitation, an actin cytoskeleton is structured between the plasma and external acrosomal membranes, which prevents the membranes from docking and the sperm from undergoing spontaneous acrosomal reaction (Breitbart et al., 2005 [doi: 10.1530/rep.1.00269.]; Ramírez-Ramírez et al., 2020 [DOI: 10.1002/jcb.29521]). This cytoskeleton is destructured when inducers of the acrosomal reaction (A23187, Progesterone or ZP3) trigger signaling pathways that elevate the intracellular concentration of calcium, allowing the activation of enzymes such as gelsolin and scinderin (Pelletier et al., 199 [doi: 10.1095/biolreprod60.5.1128.]; Finkelstein et al., 2010 [doi: 10.1074/jbc.M110.170951. ]) that fragment the actin cytoskeleton and allow the swelling of the acrosome with the consequent coupling of the plasma and external acrosomal membranes. Therefore, there should not be a significant difference in volume between non-capacitated and capacitated spermatozoa, as long as the actin cytoskeleton is normally structured.
Comment 2. In Figure 7, the authors show that the AR pattern in CAP cells is around 60%, the area (volume) measurements were performed in intact sperm alone (the remaining 40%), reacted alone or no distinction was considered.
Answer: For the assessment of acrosomal volume, only sperm that had not undergone acrosomal reaction, i.e., those that retained their acrosome, were considered. To avoid confusion, we have now included the results based on the change in acrosomal volume at 10 minutes of incubation. This time was selected because an increase in volume can be observed after 10 minutes of capacitation (Fig. 2B), and when approximately 90% of sperm retained their acrosome (Supplemental Figure 1).
Comment 3. I consider that control conditions for volume changes can potentiate the impact of the results, similar to the results of Fig. 4 (using FR180204), and Fig 6C and D using FR180204 and C3 compounds.
Answer: While inhibition of ERK1/2 by FR180204 or of RhoA by C3 completely or partially blocks actin polymerization, they appear not to affect Cl- or other ion fluxes. Therefore, it could be suggested that there would be no appreciable effect on sperm volume. However, this would need to be studied.
The next central result is shown in Fig. 4, the authors indicate that only TMEM16A inhibition affects actin cytoskeleton remodeling. However, the description of the results in Fig. 4 is not clear enough.
Comment 4. Was the fluorescence quantification performed in the acrosome region only? or in the whole cell? If the latter is true, those results can be masking the effect at the acrosome level.
Answer: Fluorescence quantification was only performed on the sperm heads. It has been clarified in section 4.7 of the methods (lines 572-573).
Comment 5. The authors discuss the relationship of TMEM16A function with the cytoskeleton remodeling process by connecting the channel's activity with signaling proteins (ERK pathway), however, it is still not clear how this can explain acrosome volume changes. Would the treatment with FR180204 induce a similar behavior as TMEM16A inhibition? Is TMEM16A localized in the acrosomal membrane? If not, how the signals are transduced from the plasma membrane to the acrosome?
Answer. We believe that TMEM16A inhibition has several important effects: the first and well-known one is the relationship between TMEM16A activity and ERK1/2 activity through the Ras/Raf/MEK signaling pathway (Wan et al., 2017. DOI 10.1186/s12943-017-0720-x), a signaling pathway present in mammalian sperm (Lamirande and Gagnon 2002 [doi: 10.1093/molher/8.2.124.], líneas 74-91). Second, our group recently reported the relationship of ERK1/2 with the activation of the GEF H1, which is related to the activation of RhoA (Salgado-Lucio et al., 20220 [doi:10.1242/jcs.239186]). Finally, TMEM16A activity participates in [Cl-]i homeostasis, such that inhibiting this calcium-dependent Cl- channel prevents Cl- effluxes (Schreiber et al., 20204. https://doi.org/10.1016/j.ceca.2024.102885), then Cl- is concentrated in the cytoplasm allowing the entry of water and, together with the absence of the actin cytoskeleton, acrosome volume is altered. On the other hand, although the inhibition of ERK1/2 by FR180204 prevents actin polymerization during capacitation, it does not prevent Cl- fluxes, so if there were an increase in cell volume, it might not be appreciable. The above has been explained in the discussion (lines 334-377).
As to whether the actin cytoskeleton is associated with the acrosomal membrane, the answer is yes, it is discussed in lines 358-375.
Comment 6. Fig. 5 shows that all the tested inhibitors produce (at different extents) the same physiological behavior, meaning an increase in the intracellular Cl- levels and a decrease in the pHi. From lines 362-364, the authors discuss that TMEM16A, CFTR, and CIC3 inhibition paradoxically causes "uncontrolled chloride flow" leading to intracellular Cl- increase. I suggest the authors discuss this more openly, citing the relevant studies along with the differences in anion conductances for each tested channel, as well as the role of intracellular localization of each protein. A schematic model would be helpful.
Answer: We have taken the reviewer's suggestion into account and analyzed the relationship between the subcellular localization of TMEM16A, CFTR, and ClC3 with the alteration of acrosome structure and volume (lines 358–377).
The relationship between the conductance of TMEM16A, CFTR, and ClC3 and Cl⁻ accumulation in the sperm cytoplasm was not included, as it requires a broad discussion that would involve not only these three channels but also other ion cotransporters such as NKCC, or antiporters as Cl⁻/HCO⁻, among others. This would result in a very extensive discussion.
Minor points:
Comment 7. In line 104, the sentence "TMEM16A, CFTR, and NPPB inhibitors were assessed" seems to indicate the name of the inhibitors and the name of the proteins inhibited by each.
Answer: The sentence "TMEM16A, CFTR, and NPPB inhibitors were assessed" was corrected to "TMEM16A, CFTR, and ClC3 inhibitors were assessed" (line 111).
Comment 8. When referring to signal intensity "dim" should be used instead of 'pale'
Answer: “pale” was changed to “dim” (lines 157 and 182).
Comment 9. In Fig. 3 (bottom) some arrows and arrowheads are shown but it is not indicated the meaning of this.
Answer: The meaning of the arrows and arrow heads are now indicated in Fig. 3 (lines 170-172).
Comment 10. The main text, says that phalloidin-FITC was employed, however, in the figure legend and the methods sections is indicated that phalloidin-TRIC was used. Clarify.
Answer: This error was corrected "phalloidin-TRIC" was changed to "phalloidin-FITC" (lines 198-200).
Comment 11. In the main text, nothing is commented about the results of FR180204
Answer: The results about the effect of FR180204 are found in lines 262-268.
Comment 12. In line 205 some numerical data is presented but there is no explanation about the units of this.
Answer: The units of this data were included (lines 214-215).
Comment 13. Lines 214-215 claim "These results suggest that Cl- fluxes are essential for maintaining intracellular Ca2+ concentration..." However, no calcium measurements were done.
Answer: This claim was changed by “These results suggest that Cl- fluxes are essential for maintaining the pHi homeostasis” (lines 223-224).
Comment 14. Figures 4A and 6A indicate that the concentration of T16Inh was 10 uM, however, other Figures say is 1 uM. Clarify if different concentrations were employed and why.
Answer: This error has been corrected in Fig. 4A and 6A. The concentration used in this study was 1 µM, which is the optimal concentration indicated by the supplier for inhibiting TMEM16A. Higher concentrations may produce undesirable side effects.
Comment 15. Fig 6B and D indicate that the units are A.U, however, this should indicate instead that it is the N/N0 ratio.
Answer: First, the WB densitometric data were analyzed using the N/N0 ratio. The data obtained from this ratio were then normalized with respect to the data from the non-capacitated spermatozoa; therefore, A.U. was used.
Comment 16. Line 327 says TMEM16F, not A.
Answer: “TMEM16F” was changed to “TMEM16A” (line 336).
Comment 17. Is the age of the animals employed considered, or only the weight?
Answer: In our experience, although guinea pigs reach sexual maturity between two and three months of age, we have found that, starting at five months of age, when they reach a weight of 800 to 900 grams, sperm production and quality are optimal for our experiments. For this reason, we use guinea pigs weighing between 800 and 900 grams. This also allows us to use the smallest number of organisms.
Comment 18. In sections 4.5, 4.7, and 4.10 the imaging conditions such as the ex/em information and objective magnification need to be included.
Answer: In sections 4.7 (lines 514-515), 4.8 (line 531), 4.9 (line 549), and 4.10 (lines 570), information about the ex/em of the fluorochromes used has been added.
Comment 19. The final Osmolarity of the media needs to be included.
Answer: The osmolarity of the Tyrode-HEPES medium was included and is similar to that reported by Rogers and Yanagimachi, 1975 (https://doi.org/10.1095/biolreprod13.5.568). Line 472.
Comment 20. Line 465 and 536 "NH4Cl" lacks the subindex.
Answer: This error was corrected, lines 494, 509, and 567.
Comment 21. The statistical analysis is not clear, no normality tests were performed to decide for parametric or non-parametric tests. Absolute p values are preferred to <0.05. The indications "* vs ** P<0.01" are not clear enough.
Answer: Although not indicated in section 4.14, the Shapiro-Wilk test was performed to determine data normality, followed by a one-way ANOVA after Tukey's analysis to compare multiple groups and a Student t-test for two groups. Only absolute values of P <0.05 were considered (lines 622-625).
Comments on the Quality of English Language
Comment 22. Language should be improved, many typos are present and seems that a direct Spanish-to-English translation was used (i.e. Fig. 6A says "capacitados" instead of capacitated, "membrane plasma' instead of plasma membrane).
Answer: The typos found have been corrected:
“membrane plasma” has been changed to “plasma membrane” (line 77).
Figure 6A has been corrected, “capacitados” has been changed to “capacitated.”
The English language has been corrected by the MDPI Editing Service, certificate No. 90548.
Round 2
Reviewer 1 Report
Comments and Suggestions for Authors
In the manuscript by Roa-Espitia and colleagues, the authors revealed previously unrecognized roles of various anion channels in sperm cell function, particularly in maintaining acrosomal integrity. heir findings offer valuable insights into the molecular mechanisms regulating sperm capacitation and simultaneously open novel and compelling avenues for future research in the field of reproductive biology.
In this revision, the authors have carefully addressed my previous comments and provided arguments supported by the existing literature about some conceptual concerns I had.
Minor points:
Several super indexes are missing (e.g. Cl- in lines 95-97, 106 in line 592, and many others in different lines)
Add the units (%) in the numerical data in lines 308, 324 and wherever necessary
I guess the acronym "ARs" in line 406 should be sAR instead
Author Response
In the manuscript by Roa-Espitia and colleagues, the authors revealed previously unrecognized roles of various anion channels in sperm cell function, particularly in maintaining acrosomal integrity. heir findings offer valuable insights into the molecular mechanisms regulating sperm capacitation and simultaneously open novel and compelling avenues for future research in the field of reproductive biology.
In this revision, the authors have carefully addressed my previous comments and provided arguments supported by the existing literature about some conceptual concerns I had.
We thank the reviewer for his insightful comments and suggestions, which have improved this manuscript. We have followed his suggestions and comments to the fullest.
Minor points:
1. Several super indexes are missing (e.g. Cl- in lines 95-97, 106 in line 592, and many others in different lines)
Answer: The manuscript was reviewed and the error corrected, Cl- was changed to Cl-.
2. Add the units (%) in the numerical data in lines 308, 324 and wherever necessary.
Answer: The manuscript was revised and units (%) were added where necessary.
3. I guess the acronym "ARs" in line 406 should be sAR instead.
Answer: The acronym ARs has been changed to sAR. Line 407.